# SL-6 Mimic Is a Biostimulant for *Chlorella sorokiniana* and Enhances the Plant Biostimulant Effect of Microalgal Extract

**DOI:** 10.3390/plants14071010

**Published:** 2025-03-24

**Authors:** Daria Gabriela Popa, Naomi Tritean, Florentina Georgescu, Carmen Lupu, Sergey Shaposhnikov, Diana Constantinescu-Aruxandei, Florin Oancea

**Affiliations:** 1Bioresource Department, Bioproducts Team, National Institute for Research & Development in Chemistry and Petrochemistry—ICECHIM, Spl. Independentei Nr. 202, Sector 6, 060021 Bucharest, Romania; daria.popa@icechim.ro (D.G.P.); naomi.tritean@icechim.ro (N.T.); carmen.lupu@icechim.ro (C.L.); 2Faculty of Biotechnologies, University of Agronomic Sciences and Veterinary Medicine of Bucharest, Blvd. Marasti Nr. 59, Sector 1, 011464 Bucharest, Romania; 3Research & Development Department, Enpro Soctech Com, Str. Elefterie Nr. 51, Sector 5, 050524 Bucharest, Romania; florentina.georgescu@enpro.ro; 4NorgenoTech, Ullernchausseén 64/66, 0379 Oslo, Norway; sas@norgenotech.no

**Keywords:** microalgal biostimulant, strigolactones, chlorophyll fluorescence, light stress, mung seedling, proton pump

## Abstract

This study aimed to evaluate the impact of a more cost-efficient strigolactone mimic SL-6 on *Chlorella sorokiniana* NIVA-CHL 176 growth in comparison with the strigolactone analog GR24 and the plant biostimulant functions of microalgal extracts. Three molar SL-6 concentrations were tested: 10^−7^ M, 10^−8^ M, and 10^−9^ M, respectively. Five parameters of microalgal growth were assessed: optical density, turbidity, biomass production, chlorophyll fluorescence, and pigment concentration. Results after 15 days of culturing revealed that the SL-6 treatments significantly enhanced biomass production (13.53% at 10^−9^ M), pigment synthesis, and photosystem II activity (14.38% at 10^−9^ M). The highest increases in pigments induced by SL-6 were 15.7% for chlorophyll *a* (at 10^−8^ M SL-6), 12.87% for chlorophyll *b* (at 10^−9^ M SL-6), 2.3% for carotenoids (at 10^−8^ M SL-6), and 10.78% for total pigments (at 10^−8^ M SL-6) per gram biomass compared to the solvent control (DMSO). Higher doses of GR24 and SL-6 (10^−7^ M) inhibited microalgal growth, reducing cell density, biomass production, and pigment synthesis. The microalgal extracts acted as plant biostimulants, stimulating root and shoot elongation and proton pump functioning of mung seedlings in the presence and absence of salt stress. The extracts from SL-6 biostimulated *C. sorokiniana* were more active as plant biostimulants than the extracts from the non-stimulated *C. sorokiniana*.

## 1. Introduction

Microalgae cultivation is more efficient in harvesting light and using mineral nutrients than terrestrial plants [1,2]. Their metabolic flexibility, i.e., ability to use organic nutrients in mixotrophic and heterotrophic modes [3,4], is exploited in bioremediation and wastewater treatment [5,6,7] and the valorization of various agro-industrial by-products, such as whey [8,9], sugar cane vinasse [10,11], wine lees [12,13,14], ethanol thin-stillage [15], hydrolyzed lignocellulosic biomass [16], or food waste [17]. Microalgal biomass is a feedstock for energy- and material-driven biorefinery systems [18,19]. Due to its high content of bioactive ingredients, microalgal biomass is a renewable resource for high-value-adding bioproducts, e.g., dietary supplements [20,21] and plant biostimulants [22,23,24].

Plant biostimulants represent a category of agricultural inputs, defined by their functions related to the improvement of “one or more of the following characteristics of the plant and/or the plant rhizosphere: (1) nutrient use efficiency, (2) tolerance resistance to (a)biotic stress, (3) quality characteristics, or (4) availability of confined nutrients in the soil or rhizosphere” [25]. Various types of extracts from microalgae demonstrated the agricultural functions specific to plant biostimulants, i.e., increased nutrient uptake and nutrient use efficiency [26], enhanced tolerance to different abiotic stress, e.g., water stress [27] or drought [28,29], improved yield quality [30], and enhanced growth and photosynthetic performance [31]. Microalgae were themselves used as terrestrial plant models to screen the bioactive components influencing photosynthesis performance, e.g., whey hydrolysate peptides [32] or hydrophobic contaminants [33].

Strigolactones (SLs) are exo- and endo-signals produced by terrestrial plants, initially identified in root exudates as inducers of seed germination in the *Striga* parasitic plant [34]. The initially discovered SLs have a complex structure, with a complex ABC-ring system linked through an enol-ether bridge to the bioactiphore, a methyl-butenolide D-ring [35]. Studies from the last decade have demonstrated a complex landscape of strigolactone signal structures. Besides the strigolactones with the complex ABC-ring (called “canonical strigolactones”), more simple molecules that maintain the bioactive D-ring (“non-canonical strigolactones”) were discovered [36]. The butenolide D-ring is essential for the interaction of strigolactones with the alpha/beta hydrolase-specific receptors [37]. The initial function of the non-canonical strigolactones that lack the complex ABC stereocenter seems to be a rhizosphere signal and later evolved into plant hormones due to root–shoot transport selectivity among strigolactones [38]. As endo-signals, SLs coordinate plant responses to (a)biotic stress [39,40] and the integration of metabolic and nutrition signals [41]. The functions of SLs as exo-signals (ecomones) are both for target organisms, i.e., as pheromones (for detection of neighboring plants) and synomones (exo-signals for arbuscular mycorrhizae fungi, nitrogen-fixing bacteria, and other plant-beneficial microbes) and for eavesdropping organisms, as kairomones (exo-signals for parasitic plants, nematodes, and fungal plant pathogens) [38,42].

Due to the difficulty of preparing natural SLs, which are complex and easily degradable structures, synthetic strigolactones were proposed [43]. The synthetic analog GR24 [44] is among the most used synthetic strigolactones for various studies demonstrating agricultural functions similar to plant biostimulants. The exogenous application of the strigolactone analog GR24 enhanced plant tolerance to drought [45], salt stress [46], heat [47], low temperature [48], and toxic elements such as cadmium (Cd) [49]. Exogenous strigolactone application increased mineral nutrient uptake, due to the stimulation of mycorrhizal symbiosis [50]. The exogenous application of GR24 improves artemisinin production in *Artemisia annua* [51]. The mechanisms involved in the plant biostimulant-like effects of exogenous strigolactone are related to enhanced photosynthetic efficiency and decreased oxidative stress due to the activation of the antioxidant systems in plants [52,53].

Although strigolactones are specific endo- and exo-signals in multicellular photosynthetic organisms, with the first appearance in the green lineage on stoneworts, liverworts, and mosses [54], the strigolactones were proved to be active on microalgae as well. The studies are nevertheless in their infancy, and not much is known about the observed effects, especially the mechanisms. 5-Deoxystrigol (5-DS), a canonical strigolactone, applied at a 10^−11^ M concentration, improved the efficiency of a co-culture *Scenedesmus obliquus*—*Ganoderma lucidum* in biogas upgrading and wastewater purification [55]. The application of the strigolactone analog GR24 was demonstrated to improve the efficiency of microalgal-based technologies for biogas upgrading and removing nutrients/pollutants from digestate. GR24, applied at a concentration of 10^−7^ M, increased the growth rate and daily productivity of *C. vulgaris* FACHB-8 cultivated in a medium that simulates biogas digestate and in the presence of biogas. This increase was due to an enhanced photosynthetic performance [56]. In microalgae, SLs have been reported to have functions similar to those exerted on terrestrial plants, an enhanced response to abiotic stress (as endo-signals), and an induction of symbiotic interactions (i.e., synomone function). SLs could be considered as “microalgal biostimulants”, a mirror term for products that have microalgal biotechnological functions analog to the agricultural functions of plant biostimulants. Some of these functions are increased nutrient use efficiency (including due to the enhanced photosynthesis performance), enhanced tolerance to abiotic stress, improved yield quality (bioactive compound accumulation in biomass), and enhanced bioavailability of confined nutrients (due to promotion of the symbioses with others organisms) [57]. Other reported “microalgal biostimulant” products are humic substances [58,59] and protein hydrolysates prepared from waste chicken feathers [60]. Microalgal biostimulants are one of the solutions to the challenges that limit microalgal-based biotechnology development.

The cost of the synthesis of strigolactone analogs is still very high. For scale-up applications, compounds that are easily accessible in significant amounts and safe are needed. Our group demonstrated that strigolactone mimics, synthetic molecules easier to synthesize, which keep only the butenolide active D-ring (more similar to the non-cano-nical natural strigolactone structure), are active on microalgae. The SL-F3 mimic, 3-(4-methyl-5-oxo-2,5-dihydrofuran-2-yl)-3*H*-benzothiazol-2-one (**7**), at a concentration of 10^−7^ M, increased the biomass accumulation due to an improved photosynthetic efficiency. The SL mimics supported microalga adaptation to the light stress determined by continuous illumination [61].

We have previously reported the synthesis of another SL mimic, SL-6, active in inducing parasitic seed germination and a modification of phytopathogenic fungi branching [62]. We improved the process for multi-gram synthesis, and we demonstrated that SL-6 has a synomone function, increasing the biofilm formation by nodulation-enhancing bacteria [63]. We recently assessed SL-6 for its ecotoxicological impact on marine and freshwater organisms, including microalgae, *Raphidocelis subcapitata*, and *Skeletonema pseudocostatum*. The estimated EC50 was 1.21 mg/L after 48 h exposure, corresponding to 0.4 × 10^−5^ M [64]. Because SL-6 can be produced in large quantities at much lower costs than SL analogs, it could be used in microalgal biotechnologies if proven to have biostimulant activity on microalgae. Therefore, this study aimed to investigate the effect of SL-6, at concentrations significantly lower than EC50, on the growth of *Chlorella sorokiniana* NIVA-CHL 17. Moreover, as the microalgal extracts were proven to have plant biostimulant effects, we hypothesized that SL-6 could enhance the biostimulant effect of *C. sorokiniana* extract. To verify this hypothesis, we tested the plant biostimulant effect on mung beans, using the extract prepared from SL-6-treated microalgae in comparison to non-treated microalgae. To the best of our knowledge, this approach of enhancing the biostimulant properties of microalgal extracts using biostimulants has not been reported before. The *C. sorokiniana* NIVA-CHL 17 extract was reported to stimulate the root elongation of *Arabidopsis thaliana* (ecotype Columbia) and the yield and photosynthetic performance of lettuce (*Lactuca sativa* L. cv. Finstar) [31]. We used GR24 as a positive control due to its demonstrated effect on microalgae from the *Chlorella* genus [65]. The microalgal extract was tested on mung seedlings in the presence and absence of salt stress to check the plant biostimulant effects. To the best of our knowledge, this is the first study to investigate (SL) biostimulated microalgal biomass as a source of extract with (enhanced) plant biostimulant functions.

## 2. Results

### 2.1. Microalgal Growth Parameters

#### 2.1.1. Optical Density

The optical density showed statistically significant effects of both the GR24 analog and SL-6 mimic on *C. sorokiniana* compared to controls (Figure 1a).

From the fifth day of culturing, marginally significant differences emerged between the control group, the SC group, and the experimental treatments with synthetic SLs, respectively. After a week of culturing, the growth rate as expressed by the optical density was significantly stimulated by c2 and c3 of GR24 and marginally significantly stimulated by the lowest concentration of mimic SL-6, c3. At the end of the culturing time, i.e., on day 15, the mean growth rates of the variants with GR24 c2 and GR24 c3 were 12.11% and 17.68%, respectively, higher compared to SC. The mean growth rates of the variants with SL-6 c2 and SL-6 c3 were 7.00% and 16.26%, respectively, higher than SC. The stimulatory effect of SL-6 on *C. sorokiniana* was slightly lower but similar to the stimulatory effect of GR24.

The highest concentration tested, c1 (1 × 10^−7^ M), of both GR24 and SL-6 induced similar or lower optical densities compared to SC, which indicates a tendency toward inhibition of the microalgal growth of applied synthetic strigolactones at higher concentrations.

#### 2.1.2. Microalgal Culture Turbidity

The turbidity measurements recorded values that followed the same trend as the optical density (Figure 1b).

The lower molar concentrations of 10^−8^ and 10^−9^ M stimulated the turbidity with statistical significance from day 5 to day 15, compared to SC. The treatments with SL-6 c2 and c3 recorded slightly lower turbidity levels than the same concentrations of GR24 but higher than controls, from day 7 to the end of the culturing period. On day 15, the treatment with SL-6, 1 × 10^−9^ M exhibited mean McFarland values that increased by 8.17% compared to SC.

The treatments with the highest dose of both strigolactones consistently exhibited the lowest turbidity values, lower than controls, a trend observed from day 5 to the end of the experiment, especially in the case of SL-6.

#### 2.1.3. Biomass Quantification

The microalgal biomass that resulted after two weeks of culturing was increased compared to control when the lower concentrations of SLs were applied (Figure 1c). Similar to previous parameters, SL-6 induced a slightly lower biomass than GR24, but it was higher than the controls. The lowest production of microalgal biomass was recorded in the variant with 10^−7^ M for both GR24 and SL-6, a little lower than the controls.

Comparing each treatment with SC, we observed distinct responses: GR24 c1 and SL-6 c1 slightly decreased the biomass by 3.96%, and 6.74%, respectively; GR24 c2 and c3 increased the biomass by 26.35% and 33.17%, respectively; SL-6 c2 and c3 increased the biomass by 8.47% and 13.53%, respectively.

The data suggest that, at optimal concentrations, both strigolactones enhance microalgal cell division and proliferation, which is crucial for biomass accumulation. These findings align with similar growth-promoting effects of some synthetic strigolactones on microalgae, reported in the literature [56,61,66].

#### 2.1.4. Extracted Pigments Concentration

Microalgal biomass is a source of valuable compounds such as pigments; therefore, we analyzed the chlorophyll and carotenoid concentrations in the cells.

The results followed the same trend as the biomass accumulation, i.e., higher dose treatments slightly inhibited pigment formation, whereas lower dose treatments stimulated chlorophyll synthesis (Figure 2).

The strigolactone analogs GR24 c2 (1 × 10^−8^ M) and c3 (1 × 10^−9^ M) induced substantial increases in ChlA compared to SC, 33.88% and 39.49%, respectively (Figure 2a). SL-6 c2 and c3 induced increases in ChlA content by 25.80% and 27.55%, respectively, relative to SC. The ChlB production was stimulated the most by the treatment with GR24 c3, with a difference of approximately 86.14% compared to SC (Figure 2b). SL-6 c2 and c3 exhibited marginally significant increases in ChlB by 6.38% and 24.81%, respectively, relative to SC.

The treatments with SL-6 c2 and SL-6 c3 increased the mean total carotenoids by 9.77% and 12.31%, respectively, compared to the solvent control, SC (Figure 2c). The same concentrations of the analog strigolactone GR24 recorded even more notable increases in total carotenoids compared to SC, with approximately 18.57% and 16.64%, respectively. SL-6 c1 (10^−7^ M) exhibited a decrease in total carotenoids by 19.00% compared to SC, whereas the same concentration of GR24 did not have an effect on the total carotenoids. The total pigments followed the same trend as its components. The lower doses, c2 and c3, of GR24 recorded significant increases in the mean total pigments compared to SC, with differences of 26.52% and 48.89%, respectively. The mimic SL-6 induced increases in the total extracted pigment quantities, by 20.08% and 24.00%, when c2 and c3 were applied, respectively, compared to SC. The highest concentration tested of both GR24 and SL-6 decreased the total pigments extracted by 8.15% but was not statistically significant (Figure 2d).

By normalizing the pigments to the microalgal biomass, one can estimate the cellular expression of the pigments. After normalization to the microalgal biomass within each treatment, the pigment content showed slightly different results compared with the non-normalized concentrations (Figure 3).

Both lowest concentrations of GR24, c2 (1 × 10^−8^ M) and c3 (1 × 10^−9^ M), induced slight increases in ChlA/g biomass compared to SC, 6.60% and 4.42%, respectively, but they were not statistically significant (Figure 3a). SL-6 c2 and c3 showed significant increases of 15.70% and 12.30% in ChlA/g biomass compared to SC, respectively (Figure 3a).

ChlB/g biomass was the highest upon the treatment with GR24 c3, with a 41.72% increase compared to SC (Figure 3b). Higher doses of GR24 recorded decreases in ChlB content per gram of microalgal biomass. SL-6 c1 and c3 showed marginally significant increases in ChlB/g biomass by 7.85% and 12.87%, respectively, relative to SC (Figure 3b).

The total carotenoids/g biomass was slightly lower than the control in the case of GR24 c3 and SL-6 c1, which showed a rather opposite behavior between the two compounds with their concentration (Figure 3c). The highest concentration of GR24 c1 (1 × 10^−7^ M) induced a small increase in carotenoids/g biomass of 4.4% compared to SC. GR24 c2 and SL-6 c2 and c3 did not show a significant increase in carotenoid expression (2.3% of SC for SL-6 c2).

The total pigments biosynthesized by the microalgal cultures per gram of biomass were the highest for the treatment with GR24 c3 (1 × 10^−9^ M), with 12.85% more than SC, followed by SL-6 c2, with 10.78% more than SC (Figure 3d). These results were statistically marginally significant.

Considering the significance of photosystem II (PSII) and the quantum yield in microalgae photosynthesis, these results provided valuable insights into the efficiency of light energy conversion in *C. sorokiniana* under different treatments (Figure 4).

Treatments with lower SL doses resulted in higher chlorophyll fluorescence values compared to the solvent control, being significant after 15 days, which suggests potentially enhanced PSII activity and improved quantum yield. This could indicate increased photosynthetic efficiency and correlates with the stimulated biomass production by these treatments. SL-6 increased the *C. sorokiniana* quantum yield by 14% when the lower doses (c2 and c3) were applied, more than GR24 (9%), after 15 days of culturing. We performed a Pearson correlation between the parameters determined above to understand how these parameters relate to each other (Figure 5).

As expected, all the parameters of the microalgal growth correlated with each other and with the pigment concentration (per liter), as the latter is a function of the total biomass. The highest growth—pigment positive correlation—was with ChlA and total pigments, and the lowest positive correlation was with ChlB. This indicates that the effect of SL on the production of pigments depends on the pigment type. A positive correlation was observed between these parameters and the chlorophyll fluorescence as well, especially for optical density, turbidity, and ChlA, which showed a statistically significant correlation.

The correlation with the normalized pigments (level of expression) presented some differences compared with the pigment concentrations. The growth parameters showed a statistically significant positive correlation only between the optical density and the total pigments/biomass. ChlA/biomass, ChlB/biomass, and total pigments/biomass correlated positively with ChlA, ChlB, and total pigments, respectively. The total carotenoids/biomass did not correlate with total carotenoids and had a negative correlation with almost all the other parameters but was not statistically significant. Other statistically significant positive correlations were between total pigments/biomass and ChlA, ChlB, and ChlB/biomass and between the chlorophyll fluorescence and ChlA/biomass and total pigments/biomass.

### 2.2. Mung Seedlings Biotests

#### 2.2.1. Effects on Seedlings Growth

The extract of the non-treated microalgal culture (CSk/CSkS) slightly increased the root (radicle) and hypocotyl length compared to the control C/CS, without any treatment, both in the absence and presence of salt (Figure 6).

The radicle length of mung seedlings was further increased significantly by treatments with extracts of microalgae stimulated by 1 × 10^−7^, 1 × 10^−8^, and 1 × 10^−9^ M SL-6, i.e., Sk-7, Sk-8, and Sk-9, respectively, compared to CSk. In the absence of salt, Sk-8 induced the highest increase in the mean radicle, 26.47% of C, whereas Sk-7 and Sk-9 induced similar increased values (approx. 17% of C). Under stress conditions (CS), the mean radicle length decreased by 16% compared to the unstressed seedlings (C). Treatments with Sk-7S, Sk-8S, and Sk-9S determined significant increases in mean radicles (29%, 23%, and 30%, respectively) compared with CS, higher than with the non-treated microalgae (CSkS, 20%). The radicles of treated seedlings reached the values of the control without salt, and even higher, indicating resilience under stress (Figure 6). The differences between the effects of the three concentrations of SL-6 were not statistically significant.

In the case of hypocotyl, CSk, Sk-7, and Sk-9 variants (in the absence of salt) had marginally significant higher mean length values (with approx. 5%) compared to the control C, and there was no statistical difference between the three treatments. Sk-8 induced a significant increase of 13.62% compared to C. When salt stress was applied, the mean hypocotyl length of CS was 42% lower compared with C. The treatments Sk-7S, Sk-8S, and Sk-9S showed significant increases (39%, 46%, and 46%, respectively) in mean hypocotyl length compared to CS, which was marginally significantly different compared to CSkS (Figure 6). The differences between the three SL-6 concentrations were not statistically significant.

The seedling length without salt stress varied between 4.19 ± 0.24 and 5.49 ± 0.12 cm among controls and treatments, with significant increases upon treatments. When salt stress was applied, the seedling length fell between 2.91 ± 0.03 and 4.25 ± 0.06 cm. Without salt stress, the longest seedlings were recorded for variant Sk-8 (27.4% higher than C) and with salt stress for variant Sk-9S (35.8% higher than CS).

The SL-6 controls did not have a significant effect on the seedling growth compared with the untreated controls (C/CS), neither in the absence (variants C-7, C-8, and C-9) nor in the presence of salt stress (variants C-7S, C-8S, and C-9S).

#### 2.2.2. Acidification of the Growth Medium

Proton pumps are integral to plant physiology, affecting nutrient uptake, growth, pH regulation, and stress responses. Acid pH induces the activation of expansins, which induces root growth [67,68]. Proton pumps extrude H^+^ ions into the soil, lowering the pH around the roots. This acidification can help in nutrient solubilization and uptake [69].

One of the most used methods to gather information on proton pumps is the medium acidification assay, which is known to provide only qualitative information. We have recently developed a semi-quantitative analysis based on this assay [70], which we used in this study as well. We estimated both the total extracellular H^+^ level (eH^+^), based on the yellow area and intensity (Figure 7c), and the specific extracellular H^+^ level (seH^+^), obtained by normalizing eH^+^ to the root area, which should reflect the proton pump activity. The CSk treatment resulted in a 13.6% increase in the eH^+^ levels compared to C, and the treatments with Sk-7 and Sk-8 exhibited 19.7% and 28.0% increases, respectively, higher eH^+^ levels than C, in the absence of salt stress. Sk-9 showed a behavior similar to CSk (Figure 7a). The controls with SL-6 (C-7, C-8, and C-9) had a small decrease in the eH^+^ levels compared to the control C, in the absence of salt.

When salt stress was applied, all four microalgal extract treatments (CSkS, Sk-7S, Sk-8S, and Sk-9S) enhanced medium acidification compared to CS in a similar trend to the no-salt situation. In particular, treatments with Sk-8S and Sk-7S had the highest effect and enhanced the eH^+^ values by 46% and 33%, respectively, compared to CS. The Sk-8S treatment brought the eH^+^ values a little higher (6%) than the control without salt (C).

The highest seH^+^ level (Figure 7b) was recorded for the treatment with Sk-9/Sk-9S, both with and without salt stress, and the values were significantly higher than the untreated controls C (by 31%), CSk (by 18%), CS (by 38%), and CSkS (by 28%), respectively. A Pearson correlation revealed that all the parameters of the mung seedlings were positively correlated with each other with high statistical significance (Figure 8).

In order to understand better the relation between the SL effects on microalgae and the microalgal extract effects on the mung seedlings, we performed a Pearson correlation between all parameters (of microalgae at 15 days and seedlings), separately for the seedlings without stress (Figure 9a) and the salt-stressed seedlings (Figure 9b).

In both cases, eH^+^ was in general positively correlated with the microalga parameters, except with ChlB/biomass and total carotenoids/biomass, i.e., with the level of synthesis of ChlB and total carotenoids. The seedling parameters had a tendency toward a negative correlation with ChlB/biomass and total carotenoids/biomass, especially in the absence of salt. The chlorophyll fluorescence parameter tends to be positively correlated with all parameters determined in mung bean seedlings, stressed or not stressed. For the salt-stressed seedlings, seH^+^ of mung roots was highly positively correlated with the chlorophyll fluorescence of the microalgae. Due to the fact that, from the eight variants of seedlings, only four (the ones treated with the extracts) could be correlated with the microalgal parameters, the positive correlation between seH^+^ and the other parameters was lost in the absence of salt and reduced in the presence of salt. The most evident cause is that seH^+^ continues to increase, whereas the other parameters decrease at the lowest SL-6 concentration (Sk-9), in the absence of salt.

## 3. Discussions

### 3.1. Lower Doses of Synthetic Strigolactones as Microalgal Biostimulant

Synthetic SLs enhance microalgal growth parameters due to an increased photosynthetic performance. The photosynthetic yield of the photosystem II, determined by recording the chlorophyll fluorescence induced by pulse amplitude-modulated (PAM) saturating light, was reported to be improved by optimal concentrations of synthetic SLs [56,61]. A similar effect of exogenously applied synthetic strigolactones on PAM-induced chlorophyll fluorescence was reported for plants submitted to abiotic stress—synthetic SLs reduced photoinhibition and improved the photosynthetic yield in plants exposed to (low) light stress [71,72] and, very recently, in microalgae [73]. In our present study, the increased photosynthetic yield (chlorophyll fluorescence) positively correlates with biomass accumulation. The correlations of the chlorophyll fluorescence with optical density and turbidity are statistically significant (Figure 5). The three parameters of microalgal growth, i.e., optical density, turbidity, and biomass, positively correlated with each other. Turbidity and optical density both depend on the number and dimension of insoluble particles, but the optical density could be influenced by the pigment color of the culture as well. The significant positive correlation between these three parameters indicates that the effects of treatment on the pigments did not significantly affect the quantification of cellular growth by optical density. The influence of GR24 and SL-6 on the microalgal growth was determined by three independent assays, which validated the results.

Chlorophyll *a* and chlorophyll *b* are pigments crucial for photosynthesis in microalgae, contributing to adaptability to various environmental conditions [67]. The positive correlation between the total biomass and pigment accumulation in the culture, ChlA, ChlA, total carotenoids, and total pigments, is (highly) statistically significant. Synthetic SLs applied at optimal concentrations (in our case, 10^−8^ M and 10^−9^ M) increased the growth rate and accumulation of photosynthetic pigments in the culture. However, the pigment content in the culture is a function of both biomass and pigment cellular synthesis; therefore, it may not reflect the evolution of pigment cellular synthesis. The cellular synthesis of the photosynthetic pigments, i.e., specific pigments per gram of biomass, tends to be influenced differently. ChlA/biomass and ChlB/biomass tend to correlate positively with biomass accumulation, and total carotenoids/biomass tends to be negatively correlated with microalgae biomass accumulation. Moreover, the cellular synthesis of carotenoids was the only one that did not correlate positively with the carotenoids in the culture (Figure 5). This results from the concentration dependence of the effects of GR24 on the synthesis of carotenoids, which is opposite to SL-6 and to all the other parameters (Figure 3c): whereas all parameters decrease or remain constant at the highest concentration (c1) both for GR24 and SL-6, the synthesis of carotenoids increases for GR24. The increase is probably determined by the drastic decrease in the cellular synthesis of ChlB at higher concentrations of GR24 (Figure 3b). The carotenoids probably became necessary in higher amounts to ensure the protection of the photosynthetic apparatus under these conditions [74].

The microalgae adapt to different light regimes by adjusting their photosynthetic pigment accumulation [75]. The carotenoid pigments are involved in the photoprotection of the thylakoids through the xanthophyll cycle [76]. In microalgal cells from the *Chlorella* genus *(C. vulgaris*), carotenoids were demonstrated to be involved in photoprotective quenching in both photosystem I and II [77] and antioxidant protection against ROS [78]. The present results suggest that there are some differences within this concentration range between the mechanisms of GR24 and SL-6, which will need more in-depth studies. We have recently shown that the effects and the concentration dependence among some SL mimics depend on the structure of SLs [61], an observation that opens new research horizons.

Applying synthetic SLs seems to compensate for light attenuation along the light path due to photosynthetic active radiation (PAR) absorption by the photosynthetic pigments of the microalgae cells from the upper layers [79,80]. Due to culture shaking, microalgal cells travel in the culture flask, from the high light at the transparent wall of flask to the low light/absence of PAR in the center of the flask culture. This situation involves transitioning from high light, which leads to photoinhibition and photosynthesis pigment loss, to light-limited conditions that reduce growth [81]. An argument for such a compensating effect of the light gradient in the microalgae culture is the growth stimulation by SL, which is more significant in cultures with a high cell density and a higher light attenuation rate. From the fifth day of culturing, marginal and later highly significant differences emerged between the control groups and the experimental treatments with synthetic SLs.

Various solutions were proposed to compensate for this drawback of microalgae cultivation at high cell density, such as applying flashing light [82,83,84], a photobioreactor design that improves homogenous light distribution, e.g., an internally illuminated photobioreactor [85,86] or a photobioreactor internally illuminated with mirror walls [87], fluorescent dyes [88], quantum dots [89], etc. Photosynthesis efficiency (determined by chlorophyll fluorescence/photosystem II yield) is increased in a microalgal culture grown in an internally illuminated photobioreactor [90] or with optimal high flashing light application [83]. Applying microalgal biostimulants based on synthetic SLs seems to represent an alternative to compensate for the stress of light attenuation in the microalgae cultivation systems. More studies are needed in this direction, including combining synthetic SLs with other solutions for homogenous light distribution and compensation of the light stress resulting from light attenuation in a high cell density culture.

The effects of synthetic SL application on endo-signal networks from microalgal cells are not yet completely understood. In *Monoraphidium* sp. QLY-1 microalgae, the application of GR24 determines increased levels of Ca^2+^ and nitric oxide (NO) acting as intracellular signals [66]. The treatment with 1 µM GR24 during the macrozooid stage of *Haematococcus pluvialis* determined a significant increase in biomass production due to increased photosynthetic efficiency. In the haematocyst stage, the same concentration of GR24 induced a higher production of astaxanthin and fatty acids, mainly due to increased biomass production [91]. In terrestrial plants, the endo-signaling of SLs interplays with Ca^2+^ [92] and NO [93]. However, endogenous SLs have not been identified in microalgae [54] despite these similarities that suggest a phytohormonal role of SLs in microalgae. The investigations completed so far have demonstrated a limited presence of the following genes coding for enzymes involved in strigolactone biosynthesis in microalgae, carotenoid cleavage dioxygenase 7 (CCD7), carotenoid cleavage dioxygenase 8 (CCD8), and 9-cis/all-trans-β-carotene isomerase (D27) [94]. Microalgal genomes are not known to include sequences coding for receptors specific for strigolactone perception and signaling, e.g., D14 (DWARF14), an α/β-hydrolase protein acting as a non-canonical strigolactone receptor, and the other proteins involved in D14-mediated SL perception, the F-box protein D3 and the D53 repressor protein [54]. The present data and previous data that show significant effects of SL and SL analogs and mimics on microalgae suggest that these molecules either mimic the activity of other molecules characteristic of microalgae or these types of hormones exist but have not been identified yet in microalgae. Either way, a receptor that responds to these molecules must exist for the cells to be affected. Whether this receptor is a specific or a non-specific one remains to be determined.

Exo-signals that include a butenolide ring were described in various biological systems, such as those isolated from lichen-derived bacteria [95] or the quorum-sensing quenchers and biofilm inhibitors produced by seaweeds/macroalgae [96]. Perception and response systems regarding such exo-signals, potentially developed in microalgae, could also detect the butenolide ring from synthetic SLs. The previously reported effects of SLs on microalgae are mainly related to the formation of association microalgae–bacteria, microalgae–fungi, or microalgae–bacteria–fungi in aquatic systems. GR24 was reported to induce the formation of different symbiotic associations that are efficient in biogas upgrading and pollutant removal. Applying 10^−9^ M GR24 increases the performance of *C. vulgaris* FACH-B—endophytic bacteria S395-2 in CO_2_ removal and consumption of organic and mineral nutrients [97]. GR24 applied at the same 10^−9^ M concentration increased the efficiency of symbiosis between *C. vulgaris* FACHB-8 and *Pleurotus geesteranus* bio-32868 and improved pollutant and CO_2_ removal [98]. 5-Deoxystrigol, applied at a concentration of 10^−11^ M, promotes the growth of the symbiotic association *C. vulgaris*–*G. lucidum*–endophytic bacteria (S395-2), biogas upgrade, and nutrient removal [99].

The induction of the tolerance to the microalgal high-density cell culture, as found in the present study, could be related to a modulation of the quorum sensing. Strigolactones modulate quorum sensing in different systems, leading to various effects. The initial function of strigolactone in terrestrial plants seems to be quorum-sensing (QS) signals. The *Physcomitrella patens* moss regulates its colony extension and protonema branching by producing strigolactones [100]. In flowering plants, SLs are also involved in detecting neighboring plants [101]. Besides these QS functions in land plants, there are similar QS activities for SLs in other biological systems. The SL analog 2′-Epi-GR24 stimulated swarming motility in rhizobia [102]. SL-6 promoted biofilm formation in *P. graminis*, a nodulation-helper bacteria [63]. Natural and synthetic strigolactones modulated quorum sensing in *Vibrio cholerae*, promoting biofilm formation and reducing toxin production [103]. The marine *V. cholerae* bacteria developed its quorum-sensing coordinated biosynthesis of human toxin during shrimp chitin hydrolysis [104]. Hydrolyzed (shrimp) chitin was reported to promote microalgae growth [105].

The treatment effects on all parameters were concentration-dependent, with lower concentrations being in general more biostimulant. It is possible that the biostimulant effect would be higher at concentrations lower than 10^−9^ M. The tendency for inhibition compared with the control at the highest concentration tested (10^−7^ M) indicates the hormetic effect of SL-6. We have previously shown that SL-6 induces significant genotoxicity on microalgae at concentrations higher than 10^−8^ M [64]. This correlates with and might explain the small inhibition observed at 10^−7^ M.

Using synthetic SLs as microalgal biostimulants has implications for applied research. Increased microalgae tolerance to abiotic stress, including light (attenuation) stress, is important for higher yield and practical applications for microalgae. Our study is the first to show that SL mimics optimized to be produced in large quantities (grams), at a cost that is almost 100 times lower than GR24, can have an effect on microalgae close to the SL analog GR24 at the same concentrations. The effect on other microalgal species of interest should be tested in order to confirm the general application of SL-6. Elucidation of the microalgal systems involved in perception and response to butenolide exo-signals is a fundamental research direction, which will lead to a better understanding of microalgae/phytoplankton ecology or lichen formation.

### 3.2. Biostimulant Effects of Microalgal Extracts Toward Mung Seedlings

The plant biostimulant functions of microalgae extracts have undoubtedly been demonstrated in the last decade [23,24,106,107]. The seed treatment with microalgae extracts improved the response of resulting seedlings and/or plants to salt stress in wheat [108], bell pepper [109], and leafy lettuce [110]. Among the mechanisms involved in salt stress mitigation by microalgae extract are enhanced root growth and improved nutrient uptake [111]. In our study, the extracts from *C. sorokiniana*, which were stimulated with different doses of SL-6 strigolactone mimic positively influenced mung seedling development and physiology in the presence or the absence of salt stress. The effects were higher than those of the non-stimulated microalgal culture. The control treatments C-7, C-8, and C-9 were included to evaluate the baseline effects of residual SL without microalgal extract. The results indicated that these controls did not significantly impact the measured parameters, including radicle length, hypocotyl length, plant height, and proton pump activity. The same observations were made when the salt stress was applied. This result suggests that the enhanced roots and shoots and proton pump activity are mainly due to the active ingredients from the microalgal extracts and not from the residual effect of SL mimic, if other active compounds are not generated from SL-6. This is especially evident for the eH^+^ and seH^+^ in the presence of salt, as the extracts of the stimulated microalgae had a very significant effect. From an application point of view, any possible contribution from residual SL does not affect the final aim of obtaining a superior biostimulant formulation more than mere microalgal extract. We plan, nevertheless, to monitor in future studies the fate of SLs upon microalgal cultivation and check for the effects of any residual SL and/or other resulting compounds detected. We should not entirely exclude the contribution from SL residues, as there could exist more complex situations such as synergic behavior, which require very detailed analysis.

The root proton pump enhances nutrient bioavailability, especially phosphorus, and nutrient uptake [69] and is a marker for biostimulant action [112]. Results obtained in our study for proton pump activity showed that the treatments with the microalgal extracts enhanced the extracellular H^+^ level observed and measured by the medium acidification (and the resulting yellow color). The same happened for seH^+^, wherein the area of the root (radicle) was considered. The chlorophyll fluorescence of microalga culture tends to be positively correlated with the root seH^+^ under normal conditions, and it is correlated at a highly significant statistical level with the root seH^+^ in salt-stressed seedlings.

This correlation suggests an involvement of the strigolactone exo-signals as a mediator of the associative interactions between edaphic (soil) benefic microalgae and plants (roots). SL-6 is a strigolactone mimic acting as exogenous signals for rhizosphere organisms [62]. Studies from the last decade spotlight the importance of microalgae as a component of the soil–plant beneficial microbiome [113,114,115]. The microalgae from the genus *Chlorella* (*Trebouxiophyceae*) have great ecological plasticity, being adapted to both aquatic and soil environments. The strain ASIB BB67, isolated from its alpine soil habitat, re-assessed as a *C. vulgaris* strain, has a tolerance to dessication greater than the strain *C. vulgaris* SAG 211-11b, isolated from an aquatic environment [116]. *C. sorokiniana* strains were isolated from edaphic environments, irrigated soils [117], and arid soils [118]. Populations/strains from this *Chlorella* genus act as plant biostimulants/biofertilizers. The filtrate of a *C. sorokiniana* strain isolated from irrigated soil promoted wheat plant growth [117]. The *C. vulgaris* strain MACC-GA0056, isolated from a sub-tropical soil and applied as a crude culture, algal extract, and culture filtrate, improved wheat germination and growth [119].

Strigolactones are a “cry for help” in the rhizosphere [120], shaping the plant-beneficial microbiome [121,122] and recruiting microorganisms that promote plant tolerance to abiotic stress [123]. Strigolactones are released as exo-signals in the rhizosphere by salt-stressed plants in the presence of microorganisms that contribute to salt stress mitigation [124]. Chlorophyll fluorescence is a biomarker of microalgal cell health and chloroplast integrity [125]. As discussed above, the synthetic SLs (exo-signals) improved the photosynthetic performance of *C. sorokiniana* (measured as chlorophyll fluorescence). The extracts of microalgae (healthier) cells, from (synthetic) strigolactone-challenged cultures, promote seedling development and enhance seedling tolerance to salt stress. The microalgae benefit from the presence of healthier plants—lower light stress at the soil surface due to the shadow effect—and root exudates as an alternative carbon source for mixotrophic/heterotrophic growth. Such potential benefic interactions between the plant roots and microalgae suggest further investigations on the role of strigolactones as exo-signals for the recruitment of beneficial microalgae by stressed plants.

There are two potential main limitations of our study that we plan to address in the future. One is the effect reported on one microalgal strain only. In order to establish if the observations are generally valid, more microalga species should be investigated. The second is related to the understanding of the mechanism and the elements contributing to the biostimulant effects, both on microalgae and plants. The main aim of the present work was to provide a proof of concept for the advantages of combining microalgal biostimulation with enhanced plant biostimulation based on extracts of those biostimulated microalgae. The investigated parameters were selected to confirm our hypothesis, and providing in-depth details will be of course a subject of significant future work, as very little information is available at the moment. Additionally, lower SL-6 concentrations will have to be tested in order to find the optimal concentration, and the results will need to be confirmed at larger scales.

## 4. Materials and Methods

### 4.1. Materials

The strigolactone analog (±) GR24 was supplied by StrigoLab (Turin, Italy). The strigolactone mimic SL-6 (2-4-methyl-5-oxo-2,5-dihydro-furan-2-yloxy)-benzo[de]isoquinoline-1,3-dione, with a molecular mass of 309.28 Da, was synthesized as previously described [63], starting from 1,8-naphthalic anhydride.

The chemicals for microalga cultivation were sodium nitrate (NaNO_3_), calcium chloride (CaCl_2_·2H_2_O), magnesium sulfate (MgSO_4_·7H_2_O), and cobalt(II) nitrate (Co(NO_3_)_2_·6H_2_O), which were obtained from Merck (Darmstadt, Germany); citric acid, dipotassium hydrogen phosphate (K_2_HPO_4_), zinc sulfate (ZnSO_4_·7H_2_O), boric acid (H_3_BO_3_), and sodium molybdate (Na_2_MoO_4_·2H_2_O) were supplied by Scharlau (Barcelona, Spain); manganese chloride (MnCl_2_·4H_2_O) and ammonium iron(III) citrate was purchased from Carl Roth (Karlsruhe, Germany), and EDTA disodium salt (Na_2_EDTA·2H_2_O) was purchased from Fluka (Honeywell, Morris Plains, NJ, USA); copper sulfate (CuSO_4_·5H_2_O) and sodium carbonate (Na_2_CO_3_) was procured from Chimopar (Bucharest, Romania). The reagent for chlorophyll extraction and SL solubilization, dimethyl sulfoxide (DMSO) was purchased from MerckDarmstadt. For the mung biotest, sodium hypochlorite solution 15% *w*/*v*, from Scharlau, sodium chloride, and ethanol from Chimreactiv (Bucharest, Romania) were used.

Agar, NaOH, and bromocresol purple, used for the proton pump bioassay of the mung seedlings, were purchased from Scharlau (Barcelona, Spain).

The strain *C. sorokiniana* NIVA-CHL 17 was supplied by the Norwegian Culture Collection of Algae, NORCCA, the Norwegian Institute for Water Research (NIVA). Mung bean seeds (*Vigna radiata* (L.) R. Wilczek var. *radiata*) were supplied by a Romanian distributor of Vilmorin (Vilmorin, La Ménitré, France).

### 4.2. Testing the Effect of SL-6 on Microalgal Growth

Both GR24 and SL-6 were first solubilized in dimethyl sulfoxide (DMSO). Two stock solutions of 1 mM were prepared, one for GR24 and one for SL-6, considering the molar mass of each strigolactone; for GR24, MWt = 298.29 g/mol, and for SL-6, MWt = 309.28 g/mol. The stock solutions were used to obtain the final tested concentrations, by serial dilutions with BG-11 medium. The experimental treatments, 1 × 10^−7^ M (c1), 1 × 10^−8^ M (c2), and a 1 × 10^−9^ M (c3) final concentration, respectively, together with a control and a solvent control (SC) with a 10^−7^ M DMSO final concentration, were assessed in triplicate. Each replicate consisted of a 100 mL Erlenmeyer flask, wherein microalgae were cultivated in the conditions described below (4.2.1). The replicates were assigned according to a randomized experimental design.

#### 4.2.1. Microalgae Cultivation

Microalgae *C. sorokiniana* NIVA-CHL 17 were grown in sterilized 100 mL Erlenmeyer flasks, which contained 50 mL BG-11 growth media [126]. The medium was sterilized in an autoclave (Panasonic) at 121° C for 15 min. The final chemical concentrations were the following: 17.6 mM NaNO_3_, 0.24 mM CaCl_2_·2H_2_O, 0.23 mM K_2_HPO_4_, 0.3 mM MgSO_4_·7H_2_O, 31 mM C_6_H_8_O_7_·H_2_O, 0.0027 mM EDTA disodium salt, 0.021 mM ammonium iron(III) citrate, 0.19 mM Na_2_CO_3_, BG-11 Trace Metals Solution 0.1% (0.046 mM H_3_BO_3_, 9 mM MnCl_2_·4H_2_O, 0.77 mM ZnSO_4_·7H_2_O, 0.17 mM Co(NO_3_)_2_·6H_2_O, and 1.6 mM Na_2_MoO_4_·2H_2_O, 0.3 mM CuSO_4_·5H_2_O).

Fresh cultures were used for the test, with the final concentration of the inoculum 1% of the *C. sorokiniana* culture at the exponential phase. The sterile media distribution and flask inoculations were completed in a microbiological hood, Bio 2 Advantage Plus (Telstar, Barcelona, Spain), to keep axenic conditions.

The experimental treatments were incubated in a growth chamber, AlgaeTron AG-230-Eco (Photon Systems Instruments, Drásov, Czech Republic), under controlled conditions of light (white, fluorescent lamp at 130 μmol/m^2^·s) and temperature (25 ± 1 °C), with a photoperiod of 14/10 h of light/dark. For 15 days, the vessels were continuously shaken by an orbital shaker, Unimax 1010 (Heidolph, Schwabach, Germany), set at 140 rpm.

#### 4.2.2. Measurement of Growth Parameters

The effect of GR24/SL-6 on *C. sorokiniana* was evaluated by measuring the main growth parameters such as the optical density, turbidity, biomass production, and pigment concentrations for 15 days.

*Optical Density*: To assess the microalgal growth, the absorbance at 750 nm of a ten-fold dilution sample was measured with a UV-Vis spectrophotometer, USB4000-UV-Vis Ocean Insight (Orlando, FL, USA). This wavelength was selected to minimize measurement errors caused by chlorophyll pigment interference [127]. The parameter was measured after 3 days, 5 days, 7 days, 11 days, and at the end of the experiment, after 15 days, respectively.

*Turbidity*: The cell density of the cultures was determined by measuring turbidity using a turbidimeter Grant-Bio DEN-1B (Grant Instruments, Shepreth, Cambridgeshire United Kingdom), expressed in McFarland units. The analysis was conducted with a working volume of 2 mL culture, harvested under aseptic conditions.

*Biomass*: For biomass determination, a volume of 5 mL from each variant was aseptically pipped in a pre-weighed Falcon tube and centrifuged at 9000× *g* for 10 min using a Universal 32 centrifuge (Hettich, Tuttlingen, Germany). After removing the supernatant, samples were dried to a constant weight in an oven, Memmert UN 75 (Memmert, Schwabach, Germany), at 50 °C [128].

*Pigment concentration*: 2 mL of each sample was centrifuged at 8000× *g* for 10 min at 20 °C, to separate the biomass from the supernatant. A total of 2 mL of DMSO pre-heated at 60 °C was added over the remaining pellet and vortexed for 10 min. Next, the eppendorf tubes were centrifuged at 8000× *g* for 10 min. The absorbance was measured at three different wavelengths, 480 nm, 649 nm, and 665 nm, necessary to calculate the content of chlorophyll *a,* chlorophyll *b*, total carotenoids, and total pigments in the samples (mg/L), according to the following formulas [128]:(1)$$\mathrm{Chlorophyll}\;a\;\left(\mathrm{ChlA}\right)=12.47\times \left({\mathrm{OD}}_{665}\right)-3.62\times \left({\mathrm{OD}}_{649}\right)$$(2)$$\mathrm{Chlorophyll}\;b\;\left(\mathrm{ChlB}\right)=25.06\times \left({\mathrm{OD}}_{649}\right)-6.5\times \left({\mathrm{OD}}_{665}\right)$$(3)$$\mathrm{Total}\;\mathrm{carotenoid}=\left[1000\times \left({\mathrm{OD}}_{480}\right)-1.29\times \left(\mathrm{ChlA}\right)-53.78\times \left(\mathrm{ChlB}\right)\right]/220$$(4)$$\mathrm{Total}\;\mathrm{pigments}=\left(1\right)+\left(2\right)+\left(3\right)$$

Additionally, the expression of pigments per gram of biomass was determined by normalizing the concentration to the microalgal biomass.

*Chlorophyll fluorescence:* The chlorophyll fluorescence was measured after 7 and 15 days of culturing, using a PAM fluorometer (PSI AquaPen AP 110/P, Photon Systems Instruments, Drásov). The manufacturer’s instructions were followed, to let the microalgal cultures adapt to dark for 10 min. The maximum quantum yield of PSII photochemistry ϕP0 was calculated according to the following formula:(5)ϕP0= Fv/Fm
where F*m*—maximum chlorophyll *a* fluorescence (after actinic flash), F*v*—maximum variable fluorescence, F*m*-F*o*; F*o*—minimum chlorophyll *a* fluorescence (after dark adaptation).

### 4.3. Preparation of Microalgal Biomass Extract

For microalgal extracts, we selected the *C. sorokiniana* biomass from the treatments stimulated by the SL-6 mimic, which is more advantageous than GR24 from the economic point of view.

The microalgal biomass was concentrated by centrifugation at 9000× *g* for 10 min, and the suspension was brought to the concentration of 0.8 g/L by removing excess volume.

The suspensions were subjected to an ultrasonication treatment for cell rupture and extraction of the intracellular active ingredients. Each suspension was ultrasonicated for 10 min, with 20 s ON and 20 s OFF time, with an 80% amplitude, using a 400 W ultrasonic processor, UP400St (Hielscher, Teltow, Germany) [129]. The extracts were centrifuged at 4 °C and 9000× *g* for 10 min, and the supernatant was collected. Solutions were filtered first by a 1.2 μm cellulose acetate filter (Sartorius Stedim Biotech GmbH, Göttingen, Germany) using a vacuum pump and, for sterility, with a 0.2 μm PES sterile syringe filter. The final extract solutions combined both the extracellular microalgal biochemicals and the extracted intracellular active ingredients.

### 4.4. Plant Biostimulant Biotests on Seedlings

Mung bean seeds (*Vigna radiata*) were selected to test the microalgal extracts. The seeds were disinfected according to the following procedure: 3 min 90% EtOH immersion, water rinse, followed by 3 min 7% sodium hypochlorite immersion and rinsing 10 times with sterile water [130]

The biotests were carried out in 90 mm sterile Petri dishes, each one containing two sterile gauze patches. On top, 9 mL of each extract was aseptically pipetted, and 10 seeds were evenly placed using a sterilized tweezer. The experimental variants included controls with water, controls with SL-6 solutions (1 × 10^−7^ M, 1 × 10^−8^ M, and 1 × 10^−9^ M, respectively), and treatments with the microalgal extracts, both without and with salt stress with a final molar concentration of 50 mM NaCl. Each variant consisted of 3 Petri dish replicates. The mung seeds were incubated in a growth chamber, Micro Clima-Series^TM^ Economic Lux Chamber (Snijders Labs, Tilburg, The Netherlands), with a temperature of 26 °C for a 16 h light cycle, 240 μmol m^−2^ s^−1^ constant led light and 22 °C for an 8 h dark cycle, and they were monitored for 5 days of growth.

#### 4.4.1. Mung Bean Seedling Measurement

At the end of the growth period, the mung bean seedlings were photographed using a professional camera. Using the ImageJ software version 1.53k [131], the root (radicle) and shoot (hypocotyl) lengths were measured.

#### 4.4.2. Medium Acidification Assay

The method applied was conducted based on our previously developed semi-quantitative assay [70]. On the second day of seedling growth, a solution of 0.04 g/L bromocresol purple in ddH_2_O was prepared and adjusted to pH 6.7 with 1 N NaOH and supplemented with agar at a final concentration of 0.75%, autoclaved at 121 °C, and poured into sterile 90 mm Petri dishes. One mung seedling of each triplicate was placed on the partially hardened agar at approximately 30 °C, pushing the roots into the agar until being partially embedded into the gel. After 24 h, the plates were photographed, and the yellow area, the color intensity of the yellow area, and the root area were measured with ImageJ. The data were used to semi-quantify the specific extracellular H^+^ level (seH^+^) and the total H^+^ level (eH^+^), using the following equations: [70](6)$$\;{\mathrm{seH}}^{+}=\frac{A_{y}\times I}{A_{r}}$$(7)$${\mathrm{eH}}^{+}=A_{y}\times I$$
where A_y_ is the yellow area, I is the intensity of the yellow area, and A_r_ is the root area.

### 4.5. Statistical Analysis and Figure Design

The data underwent one-way ANOVA analysis (with a significance level set at *p* ≤ 0.05), followed by a comparison of mean differences using the Tukey HSD post hoc test. Each experiment was conducted in triplicate (three biological samples), presenting the results as mean ± standard deviation. The graphs from the figures were designed using Origin Pro 2018 software (OriginLab, Northampton, MA, USA). IBM SPSS Statistics for Windows, software version 26.0.0.0 (IBM, Armonk, NY, USA), was used for statistical analysis (one-way ANOVA) and Pearson correlation analysis (Bivariate correlations). The Pearson correlations were performed between the microalgae parameters after 15 days of culture, between mung seedling parameters, and between both microalgae and seedling parameters (selecting the 4 variants of mung treated with microalgal extracts). Marginally significant differences were defined between samples with statistic results having common letters (e.g., ab–bc, cde–efg, etc.) and are mentioned in the Figure legends.

## 5. Conclusions

This study investigated the effects of a strigolactone analog (GR24) and a mimic (SL-6) on the growth and biomass production of *C. sorokiniana*. Further studies were conducted on the effects of microalgal culture extracts on mung bean development and growth. SL-6, a cost-effective alternative to GR24, promoted microalgae growth and enhanced photosynthetic performance. Treatment with SL-6 consistently resulted in comparable or superior outcomes in biomass production, pigment synthesis, and photosystem II yield (determined as chlorophyll fluorescence) compared to GR24 treatments. These findings highlight the potential of SL-6 as a cost-efficient yet highly effective solution for enhancing microalgal cultivation and biomass yield.

The higher dose of 10^−7^ M inhibited microalgal growth determined as cell density, biomass production, or pigment synthesis for both tested synthetic strigolactones, GR24 and SL-6. Lower doses stimulated the growth parameters of *C. sorokiniana*, enhancing up to 24.00% of the total pigment accumulation when SL-6 at 10^−9^ M was applied. These variations demonstrate a hormetic effect, with higher doses causing inhibition and lower doses having a stimulant activity.

The treatments with synthetic strigolactones, especially with SL-6, improved microalgae tolerance to the light attenuation stress in high-density culture. These results have practical implications for microalgae cultivation, offering an alternative to the solutions for compensating light stress in microalgae cultivated in agitated media.

Microalgal extracts were proved to have plant biostimulant functions, enhancing radicle elongation and overall seedling length in normal conditions and increasing mung bean tolerance to salt stress. The activity of the root proton pump, a marker for biostimulant action, is amplified by treatment with microalgae extracts. The extract from microalgae challenged with SL-6, especially the treatment with Sk-8 (extract of *C. sorokiniana* culture stimulated with 10^−8^ M SL-6), demonstrated a more pronounced plant biostimulant activity than the non-challenged culture. The results suggest a potential involvement of strigolactone exo-signals in establishing the mutual benefic associations between edaphic microalgae and plant roots.

## Figures and Tables

**Figure 1 plants-14-01010-f001:**
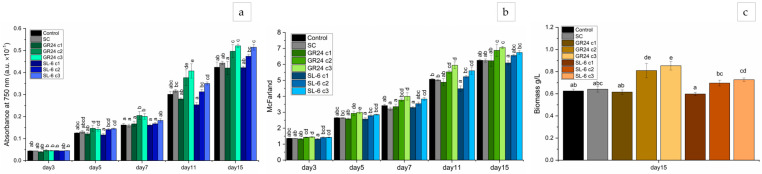
Optical density (**a**) and turbidity (**b**) during 15 days of culture; biomass after 15 days (**c**) of *C. sorokiniana* cultures. Control: growth media without treatment; SC: DMSO 0.1%, c1: 1 × 10^−7^ M, c2: 1 × 10^−8^ M, c3: 1 × 10^−9^ M. The values represent means  ±  standard deviations (n = 3 biological replicates); different letters show statistically significant differences at *p* < 0.05. Samples with common letters were considered to have marginally significant differences at *p* < 0.05.

**Figure 2 plants-14-01010-f002:**
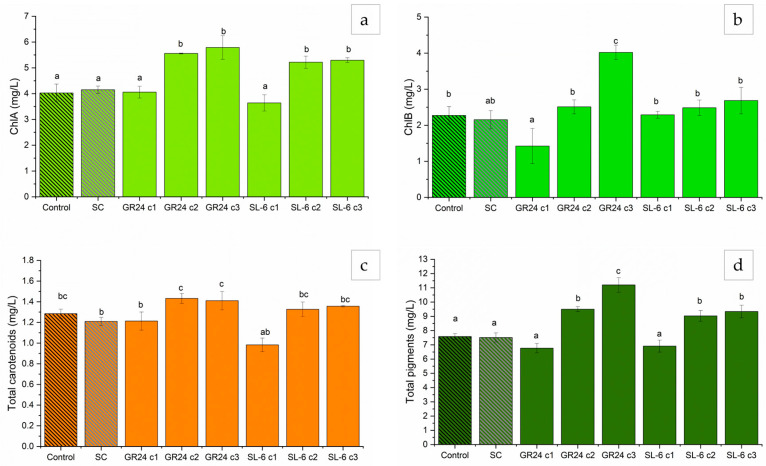
Pigment concentrations. (**a**) Chlorophyll *a* (ChlA), (**b**) Chlorophyll *b* (ChlB), (**c**) total carotenoids, and (**d**) total pigments. Control: growth media without treatment; SC: DMSO 0.1%, c1: 1 × 10^−7^ M, c2: 1 × 10^−8^ M, c3: 1 × 10^−9^. Values represent means  ±  standard deviations (n = 3 biological replicates); different letters show statistically significant differences at *p* < 0.05. Samples with common letters were considered to have marginally significant differences at *p* < 0.05.

**Figure 3 plants-14-01010-f003:**
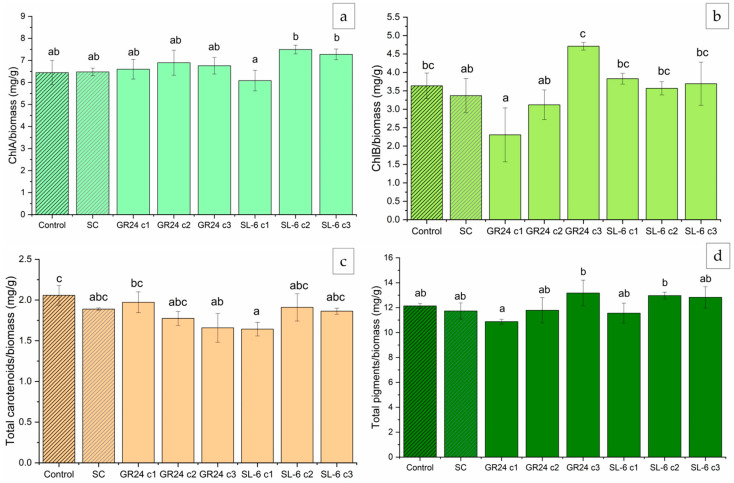
Pigment concentrations normalized to the microalgal biomass. (**a**) Chlorophyll *a* (ChlA), (**b**) Chlorophyll *b* (ChlB), (**c**) total carotenoids, and (**d**) total pigments. Control: growth media without treatment; SC: DMSO 0.1%, c1: 1 ×10^−7^ M, c2: 1 × 10^−8^ M, c3: 1 × 10^−9^ M. Values represent means  ±  standard deviations (n = 3 biological replicates); different letters show statistically significant differences at *p* < 0.05. Samples with common letters were considered to have marginally significant differences at *p* < 0.05.

**Figure 4 plants-14-01010-f004:**
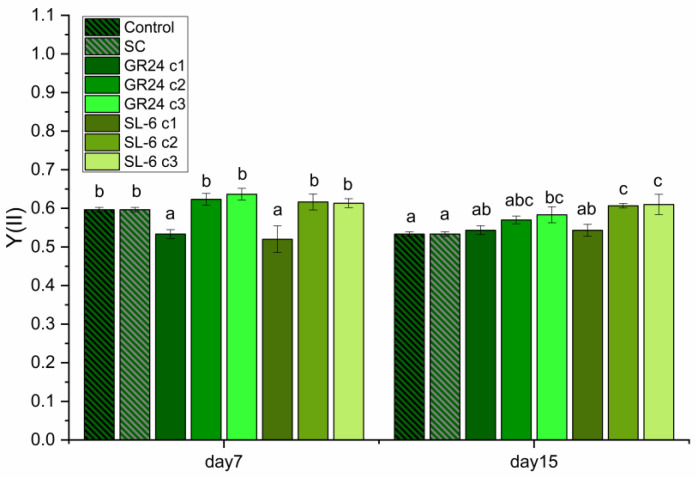
Chlorophyll fluorescence of microalgal cultures. Control: growth media without treatment; SC: DMSO 0.1%, c1: 1 ×10^−7^ M, c2: 1 × 10^−8^ M, c3: 1 × 10^−9^ M. Values represent means  ±  standard deviations (n = 3 biological replicates); different letters show statistically significant differences at *p* < 0.05. Samples with common letters were considered to have marginally significant differences at *p* < 0.05.

**Figure 5 plants-14-01010-f005:**
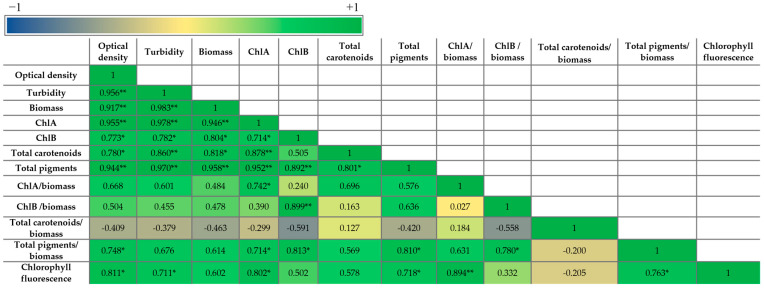
Pearson correlation between the growth parameters and pigment production of *C. sorokiniana* grown in the absence and presence of GR24 and SL-6; ** Correlation is significant at the 0.01 level; * Correlation is significant at the 0.05 level.

**Figure 6 plants-14-01010-f006:**
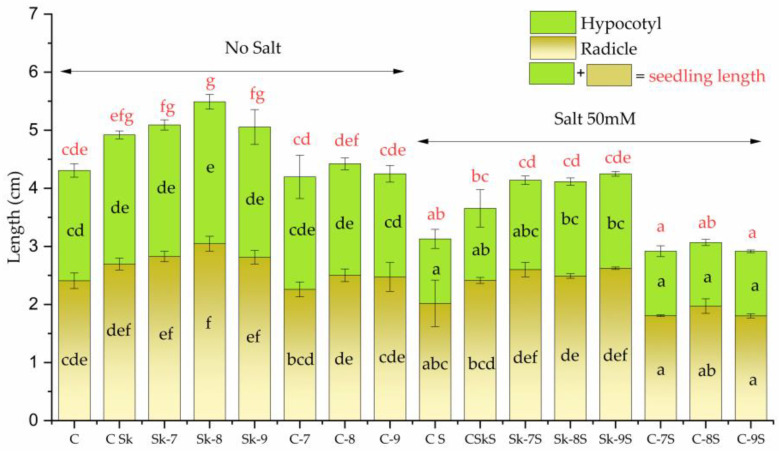
Mung seedling measurements. C, control with water; CSk, control with untreated *C. sorokiniana* biomass extract; Sk-7, extract from *C. sorokiniana* biomass treated with SL-6 c1 (1 × 10^−7^ M); Sk-8, extract from *C. sorokiniana* biomass treated with SL-6 c2 (1 × 10^−8^ M); Sk-9, extract from *C. sorokiniana* biomass treated with SL-6 c3 (1 × 10^−9^ M); C-7, control for SL-6 c1 solution; C-8, control for SL-6 c2 solution; C-9, control for SL-6 c3 solution; CS, control with water and salt stress; CSkS, control for untreated *C. sorokiniana* biomass extract with salt stress; Sk-7S, extract from *C. sorokiniana* biomass treated with SL-6 c1 (1 × 10^−7^ M) with salt stress; Sk-8S, extract from *C. sorokiniana* biomass treated with SL-6 c2 (1 × 10^−8^ M) with salt stress; Sk-9S, extract from *C. sorokiniana* biomass treated with SL-6 c3 (1 × 10^−9^ M) with salt stress; C-7S, control for SL-6 c1 solution with salt stress; C-8S, control for SL-6 c2 solution with salt stress; C-9S, control for SL-6 c3 solution with salt stress. Values represent means  ±  standard deviations (n = 3 means of 10 biological replicates). Different letters indicate statistically significant differences between samples at *p* < 0.05. Samples with common letters were considered to have marginally significant differences at *p* < 0.05.

**Figure 7 plants-14-01010-f007:**
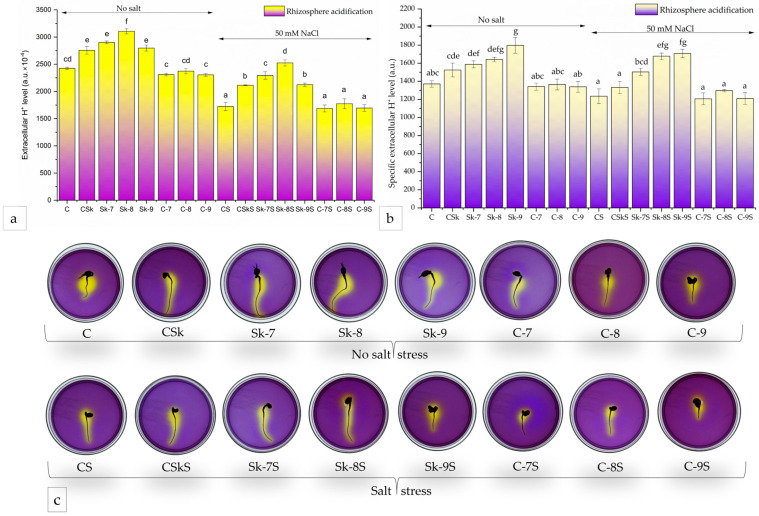
Proton pump activity; (**a**) total extracellular H^+^ level (eH^+^); (**b**) specific extracellular H^+^ level (seH^+^); (**c**) Petri dishes after medium acidification test; C, control with water; CSk, control for untreated *C. sorokiniana* biomass extract; Sk-7, extract from *C. sorokiniana* biomass treated with SL-6 c1 (1 × 10^−7^ M); Sk-8, extract from *C. sorokiniana* biomass treated with SL-6 c2 (1 × 10^−8^ M); Sk-9, extract from *C. sorokiniana* biomass treated with SL-6 c3 (1 × 10^−9^ M); C-7, control for SL-6 c1 solution; C-8, control for SL-6 c2 solution; C-9, control for SL-6 c3 solution; CS, control with water and salt stress; CSkS, control for untreated *C. sorokiniana* biomass extract with salt stress; Sk-7S, extract from *C. sorokiniana* biomass treated with SL-6 c1 (1 × 10^−7^ M) with salt stress; Sk-8S, extract from *C. sorokiniana* biomass treated with SL-6 c2 (1 × 10^−8^ M) with salt stress; Sk-9S, extract from *C. sorokiniana* biomass treated with SL-6 c3 (1 × 10^−9^ M) with salt stress; C-7S, control for SL-6 c1 solution with salt stress; C-8S, control for SL-6 c2 solution with salt stress; C-9S, control for SL-6 c3 solution with salt stress; values represent means ± standard deviations (n = 3 biological replicates). Different letters indicate statistically significant differences between samples at *p* < 0.05. Samples with common letters were considered to have marginally significant differences at *p* < 0.05.

**Figure 8 plants-14-01010-f008:**
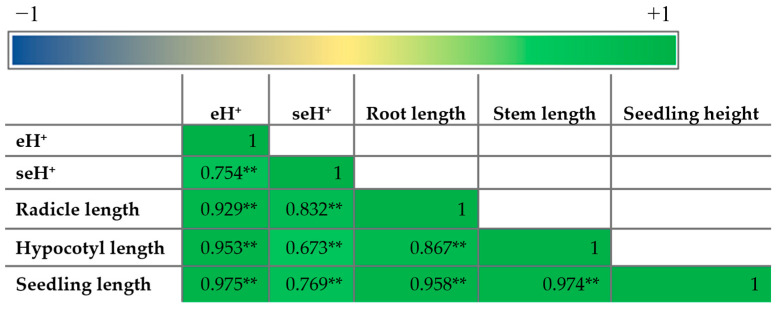
Pearson correlation between the mung seedling parameters in the absence and presence of salt stress. ** Correlation is significant at the 0.01 level.

**Figure 9 plants-14-01010-f009:**
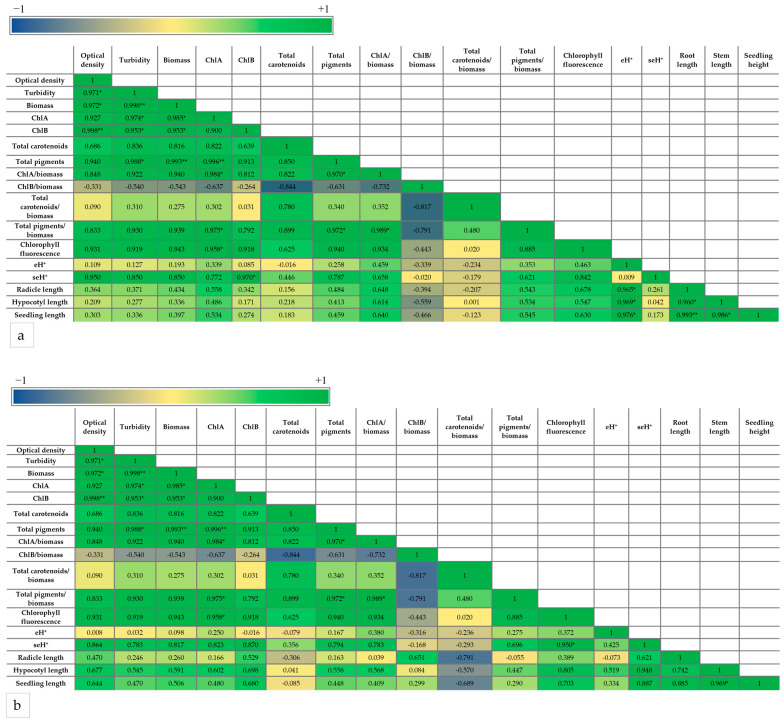
Pearson correlation between the mung seedling parameters and microalga *C. sorokiniana* parameters in the absence (**a**) and presence (**b**) of salt stress on the mung seedlings; * Correlation is significant at the 0.05 level; ** Correlation is significant at the 0.01 level.

## Data Availability

All data are included within the article.

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
