# Peer review of "SL-6 Mimic Is a Biostimulant for *Chlorella sorokiniana* and Enhances the Plant Biostimulant Effect of Microalgal Extract"

_plants, 2025, doi:10.3390/plants14071010_

Round 1
Reviewer 1 Report
Comments and Suggestions for Authors
In this study, the authors investigated the effect of SL-6 strigolactone mimic on the growth of the microalgae Chlorella sorokiniana. The extract of C. sorokiniana stimulated by SL-6 was tested as biostimulants on mung seedlings in the presence and absence of salt stress. A very interesting work that may have practical applications but also forms the basis for further scientific research in this area.
Minor revision
Lines 16-18 The sentence is unclear, rephrase it
Line 28 Add the plant on which the experiment was carried out
Lineс 275-286 Delete this section as it repeats the previously described results.
Line 579 In Materials and Methods, it states that a stock solution of GR24 and SL-6 was diluted in BG11 medium. Are the concentrations of 1 × 10-7 M (c1), 1 × 10-8 M (c2), and 1 × 10-9 M (c3) the final concentrations in the substrate in which the algae grew in the treatments?
Author Response
In this study, the authors investigated the effect of SL-6 strigolactone mimic on the growth of the microalgae Chlorella sorokiniana. The extract of C. sorokiniana stimulated by SL-6 was tested as biostimulants on mung seedlings in the presence and absence of salt stress. A very interesting work that may have practical applications but also forms the basis for further scientific research in this area.
Thank you very much for taking the time to review this manuscript. Please find the detailed responses below and the corresponding revisions/corrections highlighted/in track changes in the re-submitted files.
Comment 1: Lines 16-18 The sentence is unclear, rephrase it
Response 1: Thank you for pointing this out. We rephrased the sentence. We hope that now is clear (line 18-20).
Line 28 Add the plant on which the experiment was carried out
Response 2: Thank you for pointing this out. We added “of mung seedlings” (line 34).
Lineс 275-286 Delete this section as it repeats the previously described results.
Response 3: Thank you for this comment. We removed the section and moved the last two sentences after the previously described results. (lines 323-325 and lines 206-209).
Line 579 In Materials and Methods, it states that a stock solution of GR24 and SL-6 was diluted in BG11 medium. Are the concentrations of 1 × 10-7 M (c1), 1 × 10-8 M (c2), and 1 × 10-9 M (c3) the final concentrations in the substrate in which the algae grew in the treatments?
Response 4: Thank you for pointing this out. We added „final concentration”. (line 752).
Reviewer 2 Report
Comments and Suggestions for Authors
This study investigates the effects of SLs on microalgal growth and photosynthetic performance. The results are promising, but the main issue with the manuscript is that the data analysis is somewhat isolated and not fully aligned with the primary research objective. Additionally, the potential residue of SLs in the microalgal extracts is not addressed. It is important to consider how to avoid the residual SLs in the extracts, as this could confound the results and lead to misinterpretation. To conclude, this paper needs to revise it carefully before it can be considered. Hope the below comments will help you to further improve the paper.
Introduction
1. The paper focuses on using SLs to promote microalgal growth but only discusses the mechanisms of SLs in promoting plant growth, without addressing the mechanisms for microalgal growth promotion.
2. The paper uses SLs to promote microalgal growth and then applies microalgal extracts as biostimulants. What is the scientific basis for combining these two approaches?
3. A brief introduction to the role and importance of the butenolide active D-ring, as well as the characteristics of non-canonical strigolactones, is needed.
Materials
1. The extraction process of microalgal extracts did not remove SLs. How can their direct stimulatory effect on plant growth be avoided?
Result
4. Optical density, turbidity, and biomass are presented separately, but their correlation and biological significance are not clearly explained in the results and discussion sections.
5. At high concentrations, both SL-6 and GR24 show inhibition, but the paper doesn’t analyze if this is linked to cellular stress or toxicity.
6. The paper presents pigment content per unit biomass (mg/g) and raw data (mg/L), but does not explore the differences or correlation between them. What is the purpose of presenting both?
7. Section 3.1.4 presents extensive data but lacks focus on the experimental goal (how SLs promote microalgal growth). It should streamline non-essential parameters, emphasize concentration dependence, and clarify the mechanisms.
8. Section 3.2.1 needs further analysis on why SL-6-stimulated microalgal extracts can alleviate salt stress inhibition.
Discussion
1. The author should base the analysis and discussion on the experimental data, highlighting the innovation of the research rather than overemphasizing previous studies.
2. The limitations of the study and future research directions should be included.

Author Response
This study investigates the effects of SLs on microalgal growth and photosynthetic performance. The results are promising, but the main issue with the manuscript is that the data analysis is somewhat isolated and not fully aligned with the primary research objective. Additionally, the potential residue of SLs in the microalgal extracts is not addressed. It is important to consider how to avoid the residual SLs in the extracts, as this could confound the results and lead to misinterpretation. To conclude, this paper needs to revise it carefully before it can be considered. Hope the below comments will help you to further improve the paper.
Thank you for the observations that improve our manuscript. The primary research objective was to combine for the first time (1) the microalgal biostimulation by a synthetic SL mimic that can be produced in high amounts with (2) obtaining a plant biostimulant formulation based on the extract of the stimulated microalgae. The study does not investigate just the effects of SLs on microalgal growth and photosynthetic performance. Therefore, the paper has two main parts, part 1 about the stimulation of microalgae and part 2 about the effects of the microalgal extracts on mung seedlings. We tried to improve the data analysis and hopefully it is now more clearly presented, but please consider the observations mentioned above.
With respect to the potential residue of SLs, it is indeed a good observation. That is why we added as control in the seedling biotests the variants with just SLs (C-7/C-7S, C-8/C-8S, C-9/C-9S). We commented in the text that these controls did not induce significant effects. We added now (lines 656-657) that the difference is mainly relevant in the case of eH+ and seH+ in the presence of salt (SL alone does not have an effect, whereas the stimulated extracts have a stronger effect than non-stimulated extracts). There is the possibility that the residual concentrations might be much lower than 10-9 M (even in the case of the initial 10-7 M variant – a decrease of more than 100×) and that it could have a different effect at this concentration. For this, we nuanced now that the result “suggests” that the effect comes “mainly” from the extracts. We also added that we plan to investigate in the future the fate of SLs in the culture and test the exact residual concentration on seeds, especially that we should test on more microalgal strains in order to generalize the results. Here we would like to mention that: (i) strigolactone are hydrolyzed during their interaction with their receptors (putative and not yet known in microalgae) and (ii) the stability of strigolactone analog GR24 and strigolactone mimic SL-6 in aqueous media is limited (Halouzka et al, 2018. Stability of strigolactone analog GR24 toward nucleophiles. Pest Management Science, 74(4), 896-904; Oancea et al. 2017. New strigolactone mimics as exogenous signals for rhizosphere organisms. Molecules, 22(6), 961). In this future study we also need and want to analyze the main compounds that are affected and are responsible for the effects observed. We also want to check what the effects are in the case of harvesting the microalgae and make the extracts in water, which will completely remove any residual SL. One should also consider the production of secondary compounds that could have been formed from SL mimic during culturing. Therefore, explaining the exact mechanism and contribution from each component of the extracts needs some additional intense work. We mentioned all these at lines 659-663.
Our paper aimed to be more applied research-oriented rather than fundamental research, which also fits with the special issue to which we submitted - Microalgal Biotechnologies for Crop Production and Food Security. Considering this, finding the exact contribution to the formulated biostimulant is in fact secondary to the main objective. Here we would like to underline once again that the plant biostimulant are defined by effects (e.g., increase plant tolerance to abiotic stress) and not by the mode of action/mechanisms. The most important result is that we report a cost-effective SL mimic with which one can both stimulate the microalgae and obtain a superior plant biostimulant. We added this at lines 657-659. This is the home-take message of our paper and we tried to underline this. Hopefully we succeeded after this revision and we appreciate your valuable comments.
Introduction
- The paper focuses on using SLs to promote microalgal growth but only discusses the mechanisms of SLs in promoting plant growth, without addressing the mechanisms for microalgal growth promotion.
Response 1: Thank you for pointing this out. The reason we discuss more the mechanisms in promoting plant growth is because the mechanisms for microalgal growth promotion are almost completely unknown. SLs are well-known for their ability to orchestrate plant growth in response to environmental factors and to optime and enhance photosynthesis. However, SLs are not recognized as hormones in microalgae. There are only few published studies available that partially approach the mechanism of the effects of SLs on microalgae, as most available studies focus on the biotechnological aspects. The subject is rather new, we added this statement (lines 90-92). Nevertheless, we presented previous results and the known functions of exogenously applied SLs in microalgae to present (lines 88-109, 115-121). We added more discussion about mechanisms based on the presented data, in the Discussion section. We insisted in SLs effects on promotion plant photosynthesis, because the effects of SLs on microalgae photosynthesis that we determined in our study are analog to that on plants.
- The paper uses SLs to promote microalgal growth and then applies microalgal extracts as biostimulants. What is the scientific basis for combining these two approaches?
Response 2: Thank you for your comment. We mentioned in the paper that it is already known that the microalgal extracts have biostimulant properties (lines 638-643). We also mentioned that previous studies showed some preliminary data that SLs / SL analogs positively influence microalgal growth (lines 566 – 574), although it is not clear why and how. We recently showed preliminary results for some SL mimics (reference 61). Therefore, we thought that SL might enhance the biostimulant characteristic of the microalgae extracts as well and obtain a superior formulation. We decided to test a SL mimic previously synthesized by our group and which is promising to be used at large scale, due to its significant lower cost of synthesis.
- A brief introduction to the role and importance of the butenolide active D-ring, as well as the characteristics of non-canonical strigolactones, is needed.
Response 3: Thank you for your comment. We included a brief introduction to the role and importance of butenolide active D-ring and the characteristics of non-canonical strigolactone (lines 66-70).
Materials
- The extraction process of microalgal extracts did not remove SLs. How can their direct stimulatory effect on plant growth be avoided?
Response 3: Thank you for your comment. We partially explained in the comments above. As we already mentioned, SLs are known for their low stability in aqueous media and it is possible that the SLs are hydrolyzed during their interaction with the microalgae cellular structures, involved in the observed effects . Nevertheless, as can be seen in Figure 6 and Figure 7, we used as control the SLs without microalgal extract (C-7, C-8, C-9, C-7S, C-8S, C-9S) which did not show a significant effect. Anyway, the main aims of the paper are, on one hand to find more cost-effective SL alternatives for microalgal biostimulation (which could have additional applications such as more CO2 adsorption/fixation) and on the other hand to develop a superior plant biostimulant compared to simple SLs or microalgal extracts, which we successfully show. Therefore, avoiding the direct stimulatory effect is important for the fundamental research (to understand the mechanisms), but it is not needed for the applied research. The upscaled/industrial formulation should probably not need to remove residual SL – that are probably lower than the no effect concentration (PNEC) established for tested SL-6 (reference 64, Brooks et al 2024. An ecotoxicological assessment of a strigolactone mimic used as the active ingredient in a plant biostimulant formulation. Ecotoxicology and Environmental Safety, 275, 116244). However, as mentioned above, we plan to investigate this aspect more in depth, but several combinations are necessary to attribute any possible contribution from residual SL, if any. The exact mechanisms and contribution of the main ingredients will be the subject of future work.
Result
- Optical density, turbidity, and biomass are presented separately, but their correlation and biological significance are not clearly explained in the results and discussion sections.
Response 4: Thank you for your comments. We gathered figures 1-3 in one figure and we explained more clearly the correlation and biological significance (lines 477-484).
- At high concentrations, both SL-6 and GR24 show inhibition, but the paper doesn’t analyze if this is linked to cellular stress or toxicity.
Response 5: Thank you for the comment. The inhibition with respect to control was in fact only marginally significant from a statistical point of view, especially after 3 days of culture. We have recently shown on other microalgal species that at concentration higher than 10-8 M there is a significant genotoxicity (reference 64), so it might be related to this observation. We added a comment at lines 623-627 in this respect. We did not go into more details with respect to the small inhibition in this paper, as the mechanism of the inhibition at high concentrations was not in the scope of this particular paper. The most important aspect was to demonstrate the biostimulant effects for both microalgae and seedlings. Moreover, the study is more application-oriented and less fundamental research-oriented. We shall of course continue to investigate more in-depth the effects observed, not only on this microalgae specie, but others as well.
- The paper presents pigment content per unit biomass (mg/g) and raw data (mg/L), but does not explore the differences or correlation between them. What is the purpose of presenting both?
Response 6: Thank you for your comment. The purpose of presenting both is to characterize on one hand the pigment production or yield (mg/L) that can be obtained from the biostimulated culture (which depends on both the cellular expression of pigments and the total biomass) and on the other hand the cellular expression (mg/g). We added the following phrase “By normalizing the pigments to the microalgal biomass, one can estimate the cellular expression of the pigments” (lines 247-248). We added the correlations at lines 316-318 and discussion at lines 490-525.
- Section 3.1.4 presents extensive data but lacks focus on the experimental goal (how SLs promote microalgal growth). It should streamline non-essential parameters, emphasize concentration dependence, and clarify the mechanisms.
Response 7: Thank you for your comment. We removed non-essential parameters, emphasized concentration dependence (but in the discussion, lines 498-525, 621-627) and revised the section. Section 3.1.4 is part of the Results section and should present the results and not discuss them. The mechanism of how SL promotes microalgal growth is not known, although there are more than 10 papers available within the last 5 years, but most of them focus on biotechnological aspects rather than fundamental aspects. We added more comments in the Discussion section.
- Section 3.2.1 needs further analysis on why SL-6-stimulated microalgal extracts can alleviate salt stress inhibition.
Response 8: Thank you for your comments. Here we would like to underline once again that the plant biostimulant are defined by effects (e.g., increase plant tolerance to abiotic stress, i.e., slat stress) and not by the mode of action/mechanisms. We are not very sure what further analysis means. Analysis like interpretation of the current data or additional experiments? Interpretation does not belong in the Results section, we added more explanation in the Discussion section. The current data cannot explain more in depth the mechanism of the formulated biostimulant. More probably the microalgae extract alleviate salt stress inhibition due to a modulation of the plant response to reactive species generated by the salt stress. The investigation of the mechanism would imply testing of another hypothesis by additional investigations and should constitute a separate study/manuscript. Moreover, we repeat, the main aim of the paper was not to explain the mechanisms, we mentioned the main aims above for which we chose an applied-focused special issue. This is the first paper to show the superior biostimulant effect of SL-biostimulated microalgae and the biotechnological implications.
Moreover, if it is about mechanisms, the mechanism of the biostimulant functions is not known in general and our paper did not have the purpose of investigating in-depth the biostimulant mechanisms. The main aim of the study was to provide for the first time a proof of concept for the advantages of combining microalgal biostimulation with enhanced plant biostimulation based on extracts of those biostimulated microalgae. The biostimulant are defined by effects and not by mechanisms. We should mention here that one of the main theory in plant biostimulant science consider these biostimulants as complex systems, characterized by emergence – Yakhin et al. 2017. Biostimulants in plant science: a global perspective. Frontiers in Plant Science, 7, 2049. Biostimulant is “a formulated product of biological origin that improves plant productivity as a consequence of the novel or emergent properties of the complex of constituents, and not as a sole consequence of the presence of known essential plant nutrients, plant growth regulators, or plant protective compounds.”
We added some additional information and a new correlation between the mung parameters only which shows the high correlation between all parameters, as expected (lines 431-432, Figure 8, lines 459-464).
Discussion
- The author should base the analysis and discussion on the experimental data, highlighting the innovation of the research rather than overemphasizing previous studies.
Response 1: Thank you for your comment. We revised the discussion and highlighted the innovation of the research.
- The limitations of the study and future research directions should be included.
Response 2: Thank you for your comments. We added the limitations of the study and future research directions (lines 710-721)
Reviewer 3 Report
Comments and Suggestions for Authors
1. The title fails to fully reflect the content of the paper and contains some ambiguity.
2. The abstract is informative but could be more concise. Some details about percentages and comparisons might be better suited to the Results section.
3. The introduction is overly lengthy, with some sections delving into tangential details. For example, the historical discussion of strigolactones could be summarized. Limited justification for why this specific study is novel or necessary. Missing a clear hypothesis statement.
4. Why is the third section placed before the second section? This is quite baffling.
5. Redundancies exist between figures and textual explanations (e.g., Figures 1-4).
6. Statistical data are presented inconsistently; some results include "marginal significant differences," which could benefit from clearer definitions and explanations. But i suggest to remove the discussion on insignificant results.
7. Some microbiological names are not in italics (line 133, 152, 171).
8. Why are only the chlorophyll fluorescence parameters fv/fm measured? How about other parameters such as ΦPS II, qP, NPQ, ETR, and OJIP curves?
9. For a research article, the number of references is too large, over 130 references, even more than that of a review article.
10. The discussion often repeats results instead of integrating them with existing literature.
11. Limited critical analysis of potential limitations and alternative explanations for findings.
12. Some claims (e.g., "SL-6 promotes resilience under stress") lack direct experimental evidence and need more robust support.
13. The description of methodologies is thorough but overly detailed in parts, making it cumbersome for readers to extract key information.
14. Lack of information on how replicates and controls were handled statistically.
15. Simplify descriptions, remove redundant statements, and enhance data visualization by combining similar figures.
16. The text contains a lot of redundancies, please review the entire content for conciseness.
Author Response
- The title fails to fully reflect the content of the paper and contains some ambiguity.
Response 1: Thank you for your observation. We changed the title in order to reflect the content and results of the paper.
- The abstract is informative but could be more concise. Some details about percentages and comparisons might be better suited to the Results section.
Response 2: Thank you for your comment. We removed some data and comparisons, but left main quantitative data because it is usually standard practice, and hopefully made the abstract more concise and clearer.
- The introduction is overly lengthy, with some sections delving into tangential details. For example, the historical discussion of strigolactones could be summarized. Limited justification for why this specific study is novel or necessary. Missing a clear hypothesis statement.
Response 3: Thank you for your comment. The Introduction is approximately 2 pages (out of 20 pages of the paper, without references), so it is approximately 10% of the paper. We consider it to be appropriate. The Introduction contains the following: Importance of microalgal cultivation, which justifies the study (paragraph 1); definition of plant biostimulants and examples of microalgal extracts, to justify our approach of testing the extracts of SL-biostimulated microalgae (paragraph 2); definition and properties of SLs related to understanding the difference between SL analogs (GR24) and mimics (such as SL-6) and SL various functions (paragraph 3); known effects of GR24 on plants and the limited information of SL effects on microalgae that could help make a parallel between plants and microalgae (paragraphs 4-5); the justification of developing SL mimics due to the high cost of SL analogs (paragraph 6) and finally our previous results and the aim of this paper (paragraphs 7-8).
The tangential details are necessary because there is not much information about the SL effects on microalgae. Therefore, it is necessary to present what are the known effects on plants in order to make a parallel to the observed effects in microalgae. Moreover, the readers need to understand the difference between canonical and non-canonical SL, and between natural strigolactones, strigolactone analog and strigolactone mimics. In this respect, as requested by Reviewer 2 we added a short information on the role of the butenolide active D-ring and the properties of non-canonical SLs.
We added justification of novelty and necessity, as well as a clearer hypothesis statement (lines 129-138).
- Why is the third section placed before the second section? This is quite baffling.
Response 4: Thank you for pointing this out. The template of Plants journal is with the Material and methods section at the end. We renumbered the sections, there was an error. The order of the sections (text) was correct.
- Redundancies exist between figures and textual explanations (e.g., Figures 1-4).
Response 5: Thank you for your comment. We are not sure which redundancies is referred to. The Figures need effective, standalone captions that explain content independently from the text. The text should present the data from the figures and it should be understood separately. Moreover, in the text we presented the results as percentages (increase of decrease compared with the control, only the most important), whereas in the figures the absolute values are shown. Therefore, we cannot see any redundancies. We removed some data from the Discussion as it was repeating the data from the Results section.
- Statistical data are presented inconsistently; some results include "marginal significant differences," which could benefit from clearer definitions and explanations. But i suggest to remove the discussion on insignificant results.
Response 6: Thank you for your comment. We added the definition in Material and methods (lines 889-891) and in the Legends. We believe that the marginal significant results have some importance, at least from the point of view of showing that there is a trend. In some cases, it might not seem very significant between two concentrations, but looking at the evolution, one could say that there is a concentration / time dependence until it becomes statistically significant compared to control. This is important especially from a fundamental-oriented research approach, where we need to show that the treatment has an effect.
- Some microbiological names are not in italics (line 133, 152, 171).
Response 7: Thank you for this observation. We corrected the error and we changed to italic form.
- Why are only the chlorophyll fluorescence parameters fv/fm measured? How about other parameters such as ΦPS II, qP, NPQ, ETR, and OJIP curves?
Response 8: Thank you for your comment. The main aim of the study was to provide for the first time a proof of concept for the advantages of application of a microalgal biostimulants in the biotechnological process that lead to the preparation of an plant biostimulants with enhanced effects, based on extracts of those biostimulated microalgae. We did not go into much details with each individual mechanisms for two reasons. One was that it would have eclipsed the main message of the paper and second that it would have been too much information for one study, which already has two major parts. The parameters mentioned were not essential to prove the enhanced biostimulant effect, which was our main purpose. We can of course go later into more details once we confirm the hypothesis and select the optimal conditions. Investigation of the effects of strigolactone mimics on photosynthesis efficiency is an new hypothesis, for a new study. Before this, we should first prove that the hypothesis is still valid at larger scale and with other microalgae/plants combinations. The subject is new and opens the door for many future investigations.
- For a research article, the number of references is too large, over 130 references, even more than that of a review article.
Response 9: Thank you for your comment. We removed some references that were not essential. The relatively large number of references is justified because we have two subjects embedded in this study, one related to the SL biostimulation effect on microalgae and the second to the plant biostimulant characteristics of the extracts. We have both fundamental and applied research approach and we wanted to show the upscaling importance of the subject. Moreover, there is not enough information available with respect to this subject in order to understand the phenomena, therefore we needed to make parallels and comparisons with other known systems.
- The discussion often repeats results instead of integrating them with existing literature.
Response 10: Thank you for your comment. We removed the repeated results and improved the discussion.
- Limited critical analysis of potential limitations and alternative explanations for findings.
Response 11: Thank you for pointing this out. We added potential limitations and alternative explanations for findings
- Some claims (e.g., "SL-6 promotes resilience under stress") lack direct experimental evidence and need more robust support.
Response 12: Thank you for your comment. We could not identify this quoted claim in the text and we do not understand to which other claims the reviewer refers to. If it is related to the supposition in the discussions, that the SL-6 and GR24 effects could be related to light stress, this was not a claim, just a speculation / a mention referring to a potential mechanism. We mentioned that more studies are needed in this direction. Where direct experimental evidence was not available, we just made speculations in the discussion, which will have to be proven later.
- The description of methodologies is thorough but overly detailed in parts, making it cumbersome for readers to extract key information.
Response 13: Thank you for your comment. We removed some details that were not essential, but in some cases, it was necessary to go into detail as there are certain tricks that make the difference. For example, the best way to perform the proton pump assay, keeping the root embedded in the medium involves some important timings and procedures that are not usually explained in papers. Therefore, beginners have difficulties obtaining accurate results. We also wanted to provide all the information needed in order for others to be able to replicate our results.
- Lack of information on how replicates and controls were handled statistically.
Response 14: Thank you for pointing this out. We introduced the information on how replicates and controls were handled statistically (lines 749-754).
- Simplify descriptions, remove redundant statements, and enhance data visualization by combining similar figures.
Response 15: Thank you for your comment. We rechecked the text and we combined figures 1-3 in one figure. We revised the text.
- The text contains a lot of redundancies, please review the entire content for conciseness.
Response 16: Thank you for your comment. We revised the entire content for conciseness.
Reviewer 4 Report
Comments and Suggestions for Authors
The manuscript entitled ‘Plant biostimulant based on biomass of Chlorella sorokiniana treated with SL-6 strigolactone mimic’ aimed to investigate the effect of SL-6, at concentrations significantly lower than EC50, on the growth of C. sorokiniana NIVA-CHL 17 and the plant biostimulant effect of the extract prepared from SL-6 treated microalgae in comparison to non-treated microalgae. The findings are interesting. But there is something need to be improved before acceptance.
1. There are too many paragraphs in introduction section. Please rewrite them to increase the readability.
2. The layout of the manuscript could be enhanced for better readability. Specifically, short paragraphs at lines 326-327, 358-359, and 569-570 should be rewritten to improve coherence and flow.
3. Similar to the paragraphs, there are too many figures. I suggested the author could incorporate some similar figures as one figure.
4. In the conclusion, the reference would be better to be removed.
5. In figure 9, what does the grey part (salt 50 mM) mean? I cannot see any grey data in the figure.
6. Regarding the figure 10, how to understand some high positive correlation between specific parameters without significant difference? For example, the positive correlation between total pigment and optical density is 0.940, but they are not significant difference. What can we know about this phenomenon?
Author Response
The manuscript entitled ‘Plant biostimulant based on biomass of Chlorella sorokiniana treated with SL-6 strigolactone mimic’ aimed to investigate the effect of SL-6, at concentrations significantly lower than EC50, on the growth of C. sorokiniana NIVA-CHL 17 and the plant biostimulant effect of the extract prepared from SL-6 treated microalgae in comparison to non-treated microalgae. The findings are interesting. But there is something need to be improved before acceptance.
Thank you very much for your useful comments. Please find detailed responses below and the corresponding revisions/corrections highlighted/in track changes in the re-submitted files.
- There are too many paragraphs in introduction section. Please rewrite them to increase the readability.
R: Thank you for your comment. We made efforts to improve the Introduction section, keeping it short and easy to read while we introduced the new information (related to butenolide ring and non-canonical strigolactone) required by another Reviewer. In the Introduction we have 8 paragraphs. Each paragraph starts a new idea, as presented next, so we think that it is easier to read than having fewer and larger paragraphs. The Introduction contains the following: Importance of microalgal cultivation, which justifies the study (paragraph 1); definition of plant biostimulants and examples of microalgal extracts, to justify our approach of testing the extracts of SL-biostimulated microalgae (paragraph 2); definition and properties of SLs related to understanding the difference between SL analogs (GR24) and mimics (such as SL-6) and SL various functions (paragraph 3); known effects of GR24 on plants and the limited information of SL effects on microalgae that could help make a parallel between plants and microalgae (paragraphs 4-5); the justification of developing SL mimics due to the high cost of SL analogs (paragraph 6) and finally our previous results and the aim of this paper (paragraphs 7-8).
- The layout of the manuscript could be enhanced for better readability. Specifically, short paragraphs at lines 326-327, 358-359, and 569-570 should be rewritten to improve coherence and flow.
R: Thank you for your comment. We rewrote the mentioned paragraphs and improved the layout of the manuscript.
- Similar to the paragraphs, there are too many figures. I suggested the author could incorporate some similar figures as one figure.
R: Thank you for your comment. We combined Figures 1, 2, and 3 into one Figure.
- In the conclusion, the reference would be better to be removed.
R: Thank you for pointing this out. We removed the reference.
- In figure 9, what does the grey part (salt 50 mM) mean? I cannot see any grey data in the figure.
R: Thank you for your comment. The “grey” part was in fact a pattern for the variants with 50 mM salt, but we can understand that it is not very clear, therefore we changed the figure.
- Regarding the figure 10, how to understand some high positive correlation between specific parameters without significant difference? For example, the positive correlation between total pigment and optical density is 0.940, but they are not significant difference. What can we know about this phenomenon?
Response 6. Thank you for your comment. The strong positive correlation is not statistically significant at the p=.05 level. This means that the correlation, despite being strong in a practical sense, does not meet the threshold for statistical significance. One of the main causes of this phenomenon in our case is the contextual factors – the nature of the data, the context in which it is analyzed, the confounding variables. One example of such contextual factor influence is the strong positive correlation between total pigments and optical density without significant differences. The optical density is calculated from the transmitted light. The transmitted light is influenced by two factors: cell density (light scattering) and total pigments. Cell density determines turbidity. Optical density is statistically significantly correlated with turbidity. This means, that in our experimental conditions, the optical density statistically correlates with the number of cells and not with the total pigments. In other words, the experimental treatments determined an increase of cell growth and cell multiplication and not a (statistically) significant higher accumulation of pigments inside each microalgal cell. In Figure 5. Pigment concentrations in dependence of microalgal biomass we presented this fact. The statistical correlation underlines this effect of SL analog and mimics, on enhancing microalgal growth and multiplication. Another example - chlorophyll fluorescence is statistically correlated with chlorophyll a, and not with chlorophyll b. Chlorophyll b functions as a protection of chlorophyll a from excess light – due to its slightly different porphyrin ring, it absorbs higher energy light, from the blue-green region of the spectrum. Due to its protective function, chlorophyll b reduces chlorophyll a damage and decreases radiative dissipation of excitation energy, i.e., chlorophyll fluorescence. Therefore, in our experimental conditions, the chlorophyll fluorescence mainly results from chlorophyll a pigments from Photosynthetic System II.
Round 2
Reviewer 3 Report
Comments and Suggestions for Authors
no comments
Reviewer 4 Report
Comments and Suggestions for Authors
The author has revised the manuscript.